# Does Food Insecurity Contribute towards Depression? A Cross-Sectional Study among the Urban Elderly in Malaysia

**DOI:** 10.3390/ijerph17093118

**Published:** 2020-04-30

**Authors:** Siti Farhana Mesbah, Norhasmah Sulaiman, Zalilah Mohd Shariff, Zuriati Ibrahim

**Affiliations:** Department of Nutrition and Dietetics, Faculty of Medicine and Health Sciences, Universiti Putra Malaysia, 43400 Serdang, Selangor, Malaysia; ctfarhana.cf91@gmail.com (S.F.M.); zalilahms@upm.edu.my (Z.M.S.); zuriatiib@upm.edu.my (Z.I.)

**Keywords:** depression, food insecurity, elderly

## Abstract

With the aging of the population worldwide, there is an increasing concern for the mental health status as well as physical health. Depression is a common mental health problem among the elderly populations. Since the elderly are susceptible to food insecurity, this cross-sectional study is aimed to determine an association between food insecurity and depression among elderly people. A total of 220 elderly people- aged 60 years and above, residing in the Petaling district of Selangor, were included in this study. Face-to-face interviews were conducted to obtain the pertinent information on demographic background, food security status (six items USDA FSSM), functional status (IADL, EMS), psychosocial status (LSNS-6), and depression status (GDS-15). Binary logistic regression was used to assess the factors that were associated with depression. The median age of the elderly sample was 65.5 years. The prevalence of depression and food insecurity that was recorded were 13.2% and 19.5%, respectively. Social isolation (AOR = 5.882, 95% CI: 2.221, 15.576), food insecurity (AOR = 3.539, 95% CI: 1.350, 9.279), and unsafe mobility (AOR= 3.729, 95% CI: 1.302, 10.683) increased the odds of depression. In conclusion, social isolation, food insecurity, and unsafe mobility are factors associated with depression among the elderly people. Plans such as health interventions as well as grocery and financial aid among the qualifying elderly are suggested to improve this depression and food insecurity.

## 1. Introduction

Depression is the most common mental health problem among people of all ages worldwide. Depression can be characterised by a loss of energy, decreased interest and pleasure in usual activities, feelings of hopelessness, and complaints of memory problems [1]. Persistent depression may become a serious health problem and in severe cases can lead to suicide. Elderly people are one of the vulnerable groups who are at risk of depression. This might be due to the social and economic support systems among the elderly that have become loose due to the death of a spouse or siblings, and retirement, which result in loneliness and financial crisis, thus placing them at risk of depression.

Globally, 20% of the elderly worldwide suffer from a mental or neurological disorder, and 7% of those suffer from depression [1]. The literature review has revealed that approximately one in four elderly people has experienced depression. For instance, Bartwal, Rawat, and Awasthi (2017) reported that about 17.5% of elderly people attending primary health care in Uttarakha, India had experienced depression [2]. A local study found that 16.6% of the elderly people in Lubuk Merbau, Kedah were depressed [3]. Furthermore, according to another local study about 26.0% of the elderly people who had settled in Felda Sungai Tengi, Selangor were depressed [4]. Depression among the elderly is consistently underdiagnosed and undertreated because the symptoms may coincide with chronic disease [1].

Previous studies have documented several factors that lead to depression. Depression among the elderly could be due to their lack of involvement in social activities, older age, suffering from chronic disease, and having difficulties in falling asleep [5]. Additionally, food insecurity increases the risk of depression among the elderly [6]. Food insecurity exists whenever there is limited access to adequate and nutritious food. It is characterised by worries about food storage shortage, reduction in the frequency and quantity of daily dietary intake, or even hunger [7]. Food insecurity has been associated with poor mental health status due to the feeling of worry and anxiety about food. It is also related to cultural issues including feelings of shame, guilt, and powerlessness that is associated with food insufficiency [8].

Nevertheless, little is understood about food insecurity and depression among the elderly populations, especially among those who are living in urban areas. In Malaysia, limited studies have been conducted on food insecurity and depression among elderly people [3]. Furthermore, most previous studies on depression have been carried out in rural areas [3,4]. Therefore, this study is an important step towards determining the contribution of food insecurity to depression among the urban elderly population in Petaling District, Selangor.

## 2. Materials and Methods

### 2.1. Study and Sampling Design

This cross-sectional study was conducted to determine the contributing factors of food insecurity to depression among free-living elderly people in Petaling District. Basically, there are six subdistricts in Petaling District, namely Petaling I, Petaling II, Bukit Raja, Damansara, Sungai Buloh I, and Sungai Buloh II. Nevertheless, only two subdistricts were chosen to meet the sample size requirement, namely the Petaling II and Damansara subdistricts. These two subdistricts were randomly selected using the SPSS ‘select cases’ function. All villages in the Petaling II and Damansara subdistricts were included and elderly people aged 60 years and above in these villages were recruited. Those who were below 60 years of aged, with Alzheimer disease, or who were deaf or bedridden were automatically excluded from this study. Bedridden people were excluded as this study required the subjects to carry out several mobility tasks. A face-to-face interview based on the questionnaire was conducted to obtain the data on demographic background, food security status, functional status, and psychosocial and depression status. Demographic data that were included in this study were age, sex, marital status, education level, and monthly income. Ethics approval was approved by the Ethics Committee for Research Involving Human Subjects, Universiti Putra Malaysia (FPSK (EXP15) P014).

### 2.2. Independent Variables

#### 2.2.1. Food Security Status

The six-item U.S Department of Agriculture (USDA) Food Security Survey Module (FSSM) was used to measure the food security status among the elderly [9]. This instrument consists of affirmative responses such as ‘always’, ‘sometimes’, and ‘yes’, as well as non-affirmative responses such as ‘never’, ‘no’, and ‘don’t know’. An affirmative response was given a score of 1 and a non-affirmative response was given a 0 score. The possible total score was 6, and a score of 2 or higher indicated food insecurity. In this study, a reliability test was carried out and the Cronbach’s alpha (α) was 0.749.

#### 2.2.2. Psychosocial Status

Psychosocial status was assessed using the six item Lubben Social Network Scale-6 (LSNS-6) [10]. This scale comprises two subscales. The first subscale includes the perception of social support that is received from family and friends. The second subscale includes the number of family members and friends who are contacted by the subject to share personal problems and to seek help. This scale is a six-point scale, where 0 = none, 1 = one, 2 = two, 3 = three or four, 4 = five through eight, and 5 = nine or more. For scoring, the scale of response for each item was added in order to get the total score. Scores for each subscale ranged from 0 to 15. Hence, the total scores ranged from 0 to 30. A total score of less than 12 indicates a risk of social isolation. Based on the reliability analysis, the Cronbach’s alpha (α) for LSNS-6 was 0.616.

#### 2.2.3. Functional Status

Functional status was considered using two sub-variables, namely physical capacity and mobility status. Physical capacity was measured using the Instrumental Activity Daily Living (IADL) tool that was developed by Lawton and Brody (1969) [11]. This instrument consists of eight items including the ability to use the telephone, shopping, housekeeping, transportation, food preparation, finance handling, responsibility for own medication, and laundry. The possible total score of the IADL was 8. The ability of to carry out an activity was given a score of 1, otherwise the score was 0. A total score of less than 4 indicates physical dependence. Based on the reliability analysis, the Cronbach’s alpha (α) for the IADL scale was 0.739.

Mobility status was assessed using the Elderly Mobility Scale (EMS) that was developed by Smith (1994) [12]. The elderly need to perform all of the mobility tasks, including lying to sitting, sitting to lying, sitting to standing, standing, gait, timed walking, and functional reach. The ability to perform a task independently was given a score of 1, otherwise the score was 0. The possible total score was 20, where a score of more than 14 indicates safe mobility status. In this study, the reliability analysis showed that the Cronbach’s alpha (α) for EMS was 0.702.

### 2.3. Dependent Variable (Depression Status)

Depression status was measured using the Geriatric Depression Scale (GDS-15) that was developed by Yasavage and Sheikh (1986) [13]. This instrument consists of 15 items with yes and no responses. The possible total score was 15, where higher scores indicate a more severe levels of depression. Scores of 5 or above indicate some level of depression. A reliability analysis was carried out and the Kuder–Richardson for GDS-15 was 0.479. This scale was used to assess the depression status in this study because it had been widely used in most of the previous local studies that were related to depression status [4,14].

### 2.4. Data Analysis

All data collected were analysed using the Statistical Package for Social Science (SPSS) software version 21 (Amonk, NY: IBM Corp: 2011). For descriptive analysis, frequencies and percentages were used to present the categorical variables; meanwhile, medians and interquartile ranges (IQRs) were used to describe continuous variables as the data were not normally distributed. Chi-square test analysis was used to determine the associations between two categorical variables. The binary logistic regression enter method was used to determine the adjusted factors of depression status. Logistic regression results were described as an odds ratio (OR) with a 95% confidence interval (CI). The significance level was set at *p* < 0.05.

## 3. Results

### 3.1. Demographic Background and Food Security, Functional, Psychosocial, and Depression Status

A total of 220 elderly people had agreed to participate in this study. The median age of the sample was 65.5 years, with the majority of them (90.0%) aged 60 years to 74 years (Table 1). Over half of the sample were female (57.7%) and two thirds were married (66.8%). More than half of the sample (57.7%) had attained a primary education level. Furthermore, the median individual monthly income was RM 900, with more than half of the subjects (54.1%) having an income of below RM 940, which is the poverty line income (PLI) in Malaysia according to the Economic Planning Unit (EPU) 2014 [15]. In the context of food security status, about 19.5% of the sample were food-insecure, and 16.8% of the sample were at risk of social isolation. For functional status, about 4.5% of the elderly participants were categorised as dependent and 20.9% had an unsafe mobility status. Unsafe mobility is defined as requiring a high level of help for mobility. The prevalence of depression among the elderly subjects in this study was 13.2%.

### 3.2. Associations between Demographic Background, Food Security Status, Functional Status, and Psychosocial Status and Depression Status

Table 2 shows the associations between the demographic background, food security status, psychosocial status, and the functional status together with depression status. There were no significant associations between age, sex, marital status, educational level and monthly income and depression status (*p* > 0.05). Nevertheless, the food security status was significantly associated with the depression status (χ^2^
*=* 8.593, *p* < 0.05). The proportion of non-depressed people was higher among the food-secure (83.8%) than the food-insecure (16.2%) elderly. Likewise, fewer people were depressed among those who received high social support (86.9%) than among those who were socially isolated (13.1%). For functional status, there was no significant association between physical capacity and depression status (*p* > 0.05). However, mobility status was significantly associated with the depression status (χ^2^
*=* 7.098, *p* < 0.05), where the proportion of non-depressed people was higher among those with safe mobility (82.2%) than those with unsafe mobility (17.8%).

### 3.3. Contribution Factors of Depression

Based on the adjusted logistic regression, factors that had remained as contributing factors of the depression status were food security, psychosocial, and mobility status (*p* < 0.05) (Table 3). Food-insecure elderly people were more likely to become depressed than food-secure elderly people. Food insecurity increased the odds of depression by 3-fold (AOR = 3.539, 95% CI: 1.350, 9.279). Elderly people with low social support were almost 6 times more likely to become depressed compared with elderly people who had high social support (AOR = 5.882, 95% CI: 2.221, 15.576). Likewise, elderly people with unsafe mobility were 3 times more likely to be depressed than the elderly with unsafe mobility (AOR = 3.729, 95% CI: 1.302, 10.683). Overall, food insecurity, psychosocial factors, and unsafe mobility status contributed 23.1% of the variance in depression.

## 4. Discussion

The current study found that the prevalence of depression among the urban elderly in Petaling District was 13.2%, which was slightly lower than the prevalence of depression among the elderly in Lubuk Merbau, Kedah (16.6%) that was reported in a previous study [3]. This difference might be due to the differences in demographic and psychosocial background. Since the elderly sample lived in an urban area, the majority of the elderly people in this study reported that they were being taken care of by adult children. The urban area was the main working site for the adult children. Adult children were either living together with the elderly people or living nearby. Therefore, the elderly people in this study were surrounded by family members. Furthermore, some subjects in this study reported being involved in social activities such as religious lectures, farming, and cooking classes to fill their time and mingle with various friends. Social support might improve the ability to cope with hardships [16]. Therefore, the subjects’ emotional states were good when they were surrounded by family members and friends.

In this study, social isolation was the factor that was most strongly associated with depression. This study supported existing studies [17,18] that had identified social isolation to be a contributing factor towards depression among the elderly. Friends and family members including spouses, children, and relatives were the sources of social support. Social support is important among vulnerable groups such as the elderly. Elderly people may experience life stressors such as suffering from chronic diseases and a drop in socioeconomic status due to retirement [1]. Living alone and low social support when facing all of these stressors may contribute to poor mental and emotional health. The lack of social support is the major source of loneliness and social isolation [19]. Inversely, a high social support is associated with the increased ability to cope with stress via emotional support [17]. Hence, social activities and the presence of family members and friends might increase social support and improve the mental health status among the elderly.

Furthermore, food insecurity was associated with the risk of depression among the elderly sample in this study. This study was consistent with several previous studies [20,21] that had identified food insecurity to be associated with depression. The mechanism of this contribution can be explained by the relationship between stressors and depression [20]. Food insecurity is an example of a stressor that can be a major source of anxiety and life stress. In the psychological pathways, food insecurity leads to the shame or concern about one’s position in the social hierarchy. Financial constraints may enhance feelings of worry and anxiety about the food situation. The persistence of this stressor brings the onset of depression. On top of that, asking for food from other people and buying food on credits are considered to be socially unacceptable ways of accessing food, which creates feelings of stress, being overwhelmed, shame, and resignation. These feelings may further develop the common symptoms of mental disorders [22,23]. In addition, food insecurity is also associated with depression through a nutritional link. Existing studies have proven that food insecurity contributes to inadequate nutrient intake and deteriorating health status, such as greater risk of diseases and extended healing progress. These circumstances could also further increase the feeling of depression, especially when there is no social support network [23].

Additionally, this study revealed that unsafe mobility status was associated with depression. This study confirmed several previous studies [24,25] that described the associations between poor functional status and depression. The fear of falling is associated with functional dependency in daily living activities and depressive symptoms [25]. An inability to carry out daily living activities would decrease the pleasure that is taken in performing these usual activities. In addition, some elderly people in this study reported that they felt hopeless, suffered from chronic diseases, and poor functional status. Therefore, unsafe mobility status increased the risk of depression.

This was a cross-sectional study, which meant that a cause and effect relationship could not be drawn. In this study, only associations could be drawn between independent variables and dependent variables. Furthermore, as this study was conducted among elderly people in an urban area in Selangor, the findings cannot be generalised to other populations such as adults and children. In addition, this study included sensitive issues such as feelings of anxiety due to a shortage of food, inadequacy of food intake, and the inability to prepare a balanced food diet. Some subjects might have felt shame and denied their food insecurity situation even though their lives seemed to be full of hardships.

## 5. Conclusions

In conclusion, 13.2% of the elderly sample in this study suffered from depression. Food-insecure elderly people and elderly people with an unsafe mobility status were three times more likely to be depressed than food-secure elderly people and those with a safe mobility status. Social isolation was found to be the strongest predictor of depression among the elderly sample in this study. Health interventions such as physiotherapy and peer supports are suggested to improve the mobility and mental health of the affected elderly people. In addition, groceries and regular financial aid are proposed to decrease the prevalence of food insecurity. Family members and friends are encouraged to visit elderly people when they live nearby.

## Figures and Tables

**Table 1 ijerph-17-03118-t001:** Characteristics of the elderly.

Variables	n (%)	Median (IQR)
Demographic background:		65.5 (8.0)
Age (years)		
<75	198 (90.0)	
≥75	22 (10.0)	
Sex:		
Male	93 (42.3)	
Female	127 (57.7)	
Marital status:		
Non-married	73 (33.2)	
Married	147 (66.8)	
Education level:		
No formal education	25 (11.4)	
Primary school	127 (57.7)	
Secondary and tertiary education	66 (30.9)	
Monthly income (RM): ^1^		900.0 (1100.0)
<RM 940	119 (54.1)	
≥RM 940	101 (45.9)	
Food security status:		
Food-secure	177 (80.5)	
Food-insecure	43 (19.5)	
Psychosocial status:		
Normal	183 (83.2)	
At risk of social isolation	37 (16.8)	
Functional status:		
Physical capacity:		
Independent	210 (95.5)	
Dependent	10 (4.5)	
Mobility status:		
Safe	174 (79.1)	
Unsafe	46 (20.9)	
Depression status:		
Normal	191 (86.8)	
Depressed	29 (13.2)	

^1^ Income category based on the poverty line income (PLI) for urban areas in Peninsular Malaysia determined by the Economic Planning Unit (EPU, 2014).

**Table 2 ijerph-17-03118-t002:** Associations between demographic, food security, and functional status and depression status.

Variables	Non-Depressed n (%)	Depressed n (%)	χ^2^	*p*
Age (years):			-	0.746
<75	171 (89.5)	27 (93.1)		
≥75	20 (10.5)	2 (6.9)		
Sex:			1.239	0.266
Male	84 (44.0)	9 (31.0)		
Female	107 (56.0)	20 (69.0)		
Marital status:			1.483	0.223
Married	131 (68.6)	16 (55.2)		
Non-married	60 (31.4)	13 (44.8)		
Education level:			1.129	0.288
Below secondary level	129 (67.5)	23 (79.3)		
Secondary/tertiary level	62 (32.5)	6 (20.7)		
Monthly income (RM):			1.266	0.260
<RM 940	100 (52.4)	19 (65.5)		
≥RM 940	91 (47.6)	10 (34.5)		
Food security status:			8.590	0.003 *
Food-secure	160 (83.8)	17 (58.6)		
Food-insecure	31 (16.2)	12 (41.4)		
Psychosocial status:				0.001 *
Normal	166 (86.9)	17 (58.6)		
At risk of social isolation	25 (13.1)	12 (41.4)		
Physical capacity:			-	0.130
Independent	184 (96.3)	26 (89.7)		
Dependent	7 (3.7)	3 (10.3)		
Mobility status:			7.098	0.008 *
Safe	157 (82.2)	17 (58.6)		
Unsafe	34 (17.8)	12 (41.4)		

* Significant at *p* < 0.05.

**Table 3 ijerph-17-03118-t003:** Contributing factors to depression status.

Factors	B	Adjusted OR (95% CI)	*p*
Sex:			
Male		1.00 (ref ^1^)	
Female	0.681	1.975 (0.760 – 5.134)	0.163
Education level:			
Below secondary level	-0.014	0.986 (0.332 – 2.929)	0.979
Secondary/tertiary level		1.00 (ref ^1^)	
Monthly income (RM):			
<RM 940	0.193	1.213 (0.460 – 3.196)	0.697
≥RM 940		1.00 (ref ^1^)	
Food security status:			
Food-secure		1.00 (ref ^1^)	
Food-insecure	1.264	3.539 (1.350 – 9.279)	0.010 *
Psychosocial status:			
Normal		1.00 (ref ^1^)	
At-risk of social isolation	1.772	5.882 (2.221 – 15.576)	0.000 *
Physical capacity:			
Independent		1.00 (ref ^1^)	
Dependent	-0.606	0.546 (0.095 – 3.146)	0.498
Mobility status:			
Safe		1.00 (ref ^1^)	
Unsafe	1.316	3.729 (1.302 – 10. 683)	0.014 *

^1^ ref is a reference group, * Significant at *p* < 0.05.

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
