# Peer review of "Does Food Insecurity Contribute towards Depression? A Cross-Sectional Study among the Urban Elderly in Malaysia"

_ijerph, 2020, doi:10.3390/ijerph17093118_

Round 1

Reviewer 1 Report

This is a cross sectional study that examines the association of socio-demographic and psychological factors with depression in a sample of 220 Elderly participants with a median age of 65.5 years.

The topic addressed is worthy of investigation. Interestingly, it examines the matter of food security status in this regard, a very important and commonly neglected aspect.

The Methodology is correctly designed and described with enough detail to understand the procedures adopted. The sample of participants is adequately described for the purposes of this work.

The study finds mainly that social isolation, food insecurity and unsafe mobility are associated with depression.

The results, correctly discussed, are regarded as possibly not generalizable, and the limitations have been acknowledged in the discussion.

Overall, the results are in agreement with the current knowledge of the contributing psychosocial factors to depression.

Some drawbacks may be better explained or corrected to improve the manuscript:

  • Including some information about the concept of food insecurity and how it relates with depression would be very interesting for the readers, and would underscore the rationale for this study. It is a very important (and commonly neglected) factor in the study of nutritional psychiatry, and an asset of this study.
  • This cross-sectional methodology does not allow inferring causality relationships (inverse causality cannot be ruled out). Indeed, the factors that this study finds associated with depression very well may be caused (al least in some degree) or worsened by depression itself. Although this limitation is mentioned in page 8, the idea of a causal relationship is implicit in this manuscript and should be avoided (other than speculatively). There are some expressions throughout the text that imply causality, such as: “In conclusion, social isolation, food insecurity and unsafe mobility are strongest predictors of depression among elderly” (abstract), “social isolation was the strongest contribution factor of depression” (p. 7, line 176), “food insecurity increased the risk of the depression among elderly in this study” (p. 7, line 183), “this study revealed that unsafe mobility status was a contribution factor of depression” (p. 7, lines 191-192).

All these sentences should be rephrased to express just associations, not prediction in the sense of causality or risk. Of course, it may be hypothesized, in the discussion, that these factors may contribute to the risk of depression, but the discussion should include also how the depression may indeed cause or worsen these factors.

  • Please check the manuscript for grammatical issues.

Reviewer 2 Report

General comments:

The authors report on a cross-sectional study to assess the factors contributing to depression in urban elderly in Malaysia, thereby focusing on food security. They found a significant association between the degree of food insecurity and depression and discuss possible mediators of this association in the light of findings from other authors from other areas. The paper is well written, the results are depicted clearly and the conclusions drawn appear justified.

An overall English language editing seems advisable, and there are a few issues the authors should deal with.

Specific comments:

1.     Abstract, line 13: The wording "mental health status instead of physical health" may be understood as saying that there is now little concern about physical health. I do not think that this is true. Do you mean that there is now increasing concern over the mental health status, more than over physical health?

2.     Abstract, line 14-15: According to the title and the phrasing of the research question at the end of the Introduction (lines 51-52) the focus of this work is on the contribution of food insecurity to depression in the elderly. This is not expressed clearly in the phrasing of the research question here in the abstract.

3.     Introduction, line 43: Here it reads: "Depression among the elderly was always being undiagnosed …" Should not this read "underdiagnosed"?

4.     Materials and Methods, line 60: Here the authors say that elderly subjects were recruited in villages. The title of the paper says this was a study in urban elderly. The term "urban" points to towns/cities rather than villages. This seems to be a contradiction. In fact, there are a number of towns/cities of considerable size across Petaling district. Please clarify.

5.     Independent Variables, Food Security Status, line 71: Should this read "Food Security Survey Module"? Please check.

6.     References: I cannot find reference #3 (Rohida et al. 2017) in the journal referred to.

7.     References: I cannot find reference #4 at the location indicated.

Minor issues:

8.     Independent Variables, Psychosocial status, line 78: Typing error: Lubben Social Network Scale-6.

9.     Discussion: There is a bunch of interesting papers out on food insecurity and mental health, including depression. Since the focus of the paper is on food insecurity, I wonder if the discussion could not be extended by one ore two more interesting aspects of food insecurity and mental health. This is not a must, but could make the paper more powerful.

10.  References are not in line with the journal style: only journal name, publication year and pages are provided, but volume number is missing.
